# The crystal structure of *Thermus thermophilus* UMP kinase complexed with a phosphoryl group acceptor and donor

Kenji Fukui[1], Anzu Nishiwaki[2], Noriko Nakagawa[3], Seiki Kuramitsu[3], Ryoji Masui [2]*

1 Department of Biochemistry, Faculty of Medicine, Osaka Medical and Pharmaceutical University, Takatsuki, Osaka, Japan, 2 Graduate School of Science, Osaka Metropolitan University, Osaka, Japan, 3 Department of Biological Sciences, Graduate School of Sciences, Osaka University, Toyonaka, Osaka, Japan

* rmasui@omu.ac.jp

## Abstract

Nucleoside monophosphate kinases play crucial roles in biosynthesis and regeneration of nucleotides. Prokaryotic UMP kinase belongs to a family of amino acid kinases but not to other nucleoside monophosphate kinases. Although many structures of prokaryotic UMP kinase have been determined, limited structural information has been available on the conformational changes along the reaction and allosteric pathways. We determined the crystal structure of UMP kinase of an extreme thermophile *Thermus thermophilus* HB8 in ADP-UDP–bound form at 2.6-Å resolution. The structure of the ADP-UDP complex is the first structure of bacterial UMP kinase with a phosphoryl group donor and an acceptor. Upon simultaneous binding of ADP and UDP, the loop near ADP moved toward the active site without global open-closed conformational changes, compared to the ligand-free and UDP-bound forms. Such a shift was not observed for archaeal UMP kinases but had some similarities to those in other amino acid kinase families of enzymes.

## Introduction

Nucleoside monophosphate (NMP) kinases are ubiquitous enzymes in the biosynthesis and regeneration of ribonucleoside and deoxyribonucleoside triphosphates. Because these nucleotides are the building blocks of nucleic acids [1], enzymes responsible for their synthesis or interconversion have been targets for antibacterial drug development. NMP kinases convert respective NMPs to the corresponding nucleoside diphosphates. NMP kinases are bi-substrate enzymes that bind the two substrates simultaneously to catalyze phosphoryl transfers. Most NMP kinases are homologous families of proteins that are structurally similar. A fascinating exception to this is a prokaryotic UMP kinase (UMPK). In eukaryotes, phosphorylation of UMP and CMP is carried out by a single enzyme, UMP/CMP kinase, which resembles

**Data availability statement:** The three-dimensional structure file (known as a PDB file) is available from the Protein Data Bank database (accession numbers (PDB code) 8YH1).

**Funding:** The author(s) received no specific funding for this work.

**Competing interests:** The authors have declared that no competing interests exist.

adenylate kinase. In contrast, archaea and bacteria contain separate CMP and UMP kinases.

Prokaryotic UMPKs have no sequence similarity to other known NMP kinases but have significant sequence and structural similarity to amino acid kinases including aspartokinases and glutamate kinases [2,3]. Prokaryotic UMPKs are homohexameric enzymes. Several crystal structures of prokaryotic UMPK are known. In archaea, the crystal structures of UMPK complexed with UMP, UMP-AMPPNP and UTP have been determined for *Pyrococcus furiosu* [4]. Sructures complexed with AMPPNP and UMP-AMPPNP have also been determined for *Sulfolobus solfataricus* [5]. Additionally, in bacteria, many crystal structures of UMPK from *Escherichia coli* and other organisms have been reported and deposited: a UMP-bound form from *E. coli* [6]; UDP-bound forms from *E. coli* [6], *Helicobacter pylori* [7] and *Mycobacterium tuberculosis* [8]; a UTP-bound form from *E. coli* [6]; and ATP-bound form from *Bacillus anthracis* [9]. Archaeal and bacterial UMPKs share an α/β-fold with a β-sheet surrounded by α-helices and one $3_{10}$ helix.

However, there are some notable differences in loops surrounding the active site between archaeal and bacterial UMPKs. In archaeal UMPKs, one long flexible loop adopts various conformations depending on the presence of UMP and moves to enclose the active site in the AMPPNP-UMP complex [4,5]. This loop is specific to archaeal UMPKs and absent from bacterial UMPKs. Compared with archaeal UMPKs, only limited structural information has been available on the conformational changes in bacterial UMPKs. There is no report of a crystal structure of a ternary complex of bacterial UMPKs with a phosphoryl group donor and acceptor. More structural information is required to fully understand how ligand binding is coupled with conformational change in the active site of bacterial UMPKs.

Furthermore, bacterial and archaeal UMPKs have different regulation mechanisms [4,6,10,11]. Generally, the activity of bacterial UMPKs is stimulated by GTP, whereas that of archaeal UMPKs is not. However, there are some exceptions in bacteria: GTP has no effect on UMPK from *Ureaplasma parvum* [12]. Even more complicated, GTP binds outside the active site (allosteric site), but the *E. coli* UMPK binds GTP [13] at a location near but distinct from the binding site in *H. pylori* [7], *M. tuberculosis* [8] and *Xanthomonas campestris* [14].

In this study, we determined structures of the ADP-UDP-bound forms of UMPK from an extremely thermophilic eubacterium, *Thermus thermophilus* HB8 (ttUMPK). This organism is a gram-negative, aerobic, and rod-shaped eubacterium [15]. Because *T. thermophilus* has an optimal growth temperature of 70°C, the proteins produced in this bacterium are very stable and therefore suitable for structural and functional analyses [16,17]. Among *T. thermophilus* NMP kinases, the crystal structures of CMP kinase in various liganded states have been determined [18]. The apo form structure of AMP kinase has also been determined (PDB ID, 3 CM0). In this study, we report the first structure of bacterial UMPK with two nucleotides, a phosphoryl group donor and an acceptor. Based on these structures, we speculated the detailed conformational changes upon binding of ligands in the reaction pathway of bacterial UMPKs along with archaeal UMPKs.

## Materials and methods

### Protein overexpression and purification

The expression plasmid pET-11a/*ttumpk* (*ttha0859*) was transformed into *E. coli* BL21(DE3). This plasmid was constructed by ligating the amplified *ttha0859* fragment into the NdeI and BamHI sites of pET-11a (Merck Millipore) and provided by RIKEN BioResource Center (Tsukuba, Japan). The transformant was cultured at 310 K in LB medium containing 1 mM ampicillin, and the protein was overexpressed by induction of isopropyl β-D-thiogalactopyranoside. The cells (14 g) were harvested by centrifugation and were ultrasonicated in 20 mM Tris-HCl, pH 8.0, and 50 mM NaCl, and the lysate was incubated at 343 K for 10 min. After centrifugation, the protein in the supernatant was purified by successive chromatographic steps with RESOURCE ISO, HiTrap Heparin, RESOURCE Q and HiLoad 16/60 Superdex 75 pg columns (GE Healthcare). The purified protein (27 mg) was stored in 20 mM Tris-HCl, pH 8.0, 1 mM dithiothreitol and 150 mM NaCl at 4°C.

### Crystallization, data collection and determination

The ttUMPK (8.3 mg/mL) was incubated with 10 mM ADP, 10 mM UDP and 10 mM $MgCl_2$ in a buffer containing 20 mM Tris-HCl, pH 8.0, 150 mM NaCl and 1 mM dithiothreitol. The protein solution was diluted 1.25-fold, and 0.8 µL of the protein solution was mixed with an equal volume of a crystallization buffer containing 0.1 M sodium citrate, pH 5.6, 2.0 M ammonium sulfate and 1.0 M $LiSO_4$. Then, it was equilibrated against 100 µL of the crystallization buffer using the sitting-drop vapor diffusion method at 20°C. The obtained crystal was soaked into a crystallization buffer containing 20% glycerol and flash-cooled to 90 K in a nitrogen gas stream. Diffraction data were collected at RIKEN Harima Structural Genomics Beamline BL26B2 [19] at SPring-8 (Hyogo, Japan).

Data were processed using the HKL2000 program suite [20]. The structure of ttUMPK was solved by molecular replacement using the Phaser program in the PHENIX suite [21]. A search model for the molecular replacement was generated by prediction with the AlphaFold2 [22]. The model refinements were performed by the Coot [23] and PHENIX refine programs. The final models were validated using the MolProbity program [24]. Table 1 represents statistics for data collection and refinement.

### Sequence and structural data analysis

Sequence logos were generated with Weblogo3 [25] for the 50 most diverse members of AAK_UMPK-PyrH-Ec in the Conserved Domain Database [26]. Structure diagrams were drawn using the program PyMOL (https://pymol.org/2/).

### Enzyme assay

The activity of ttUMPK was measured at 25°C by an enzyme-coupled spectrophotometric assay [27]. The change in absorbance at 340 nm was measured with a Hitachi spectrophotometer, model U-3000. UMP (0–200 µM) was reacted with 0.1 µM ttUMPK in 50 mM Tris-HCl (pH 7.5), 100 mM KCl, 1 mM ATP, 5 mM $MgCl_2$, 0.3 mM phosphoenolpyruvate, 0.15 mM NADH, 5 units/mL pyruvate kinase (Sigma-Aldrich) and 5 units/mL lactate dehydrogenase (TOYOBO) in the presence or absence of 0.5 mM GTP. The kinetic parameters were determined using the Michaelis–Menten equation.

## Results and discussion

### Overall structure

First, we tried to determine the crystal structure of ttUMPK by molecular replacement using those of other bacterial UMPK as a search model, but phase determination was unsuccessful. Then, we used the structural models predicted by the SWISS-MODEL [28] and I-TASSER servers [29] as a search model, but this approach also failed. Finally, we succeeded in determining the UMPK structure at 2.6-Å resolution by molecular replacement using the structural model

**Table 1. Data collection and refinement statistics for ttUMPK.**

| Data collection | |
|---|---|
| Beamline | SPring-8 BL26B2 |
| Detector | RIGAKU RAXIS V |
| Wavelength (Å) | 1.0000 |
| Exposure time (s) | 1.0 |
| Camera distance (mm) | 180 |
| Oscillation angle (°) | 0.2 |
| Oscillation range (°) | 360 |
| Space group | $I222$ |
| Cell dimensions | |
| $a, b, c$ (Å) | 150.2, 232.2, 284.8 |
| α, β, γ (°) | 90.0, 90.0, 90.0 |
| Resolution (Å) | 26.32-2.60 (2.70–2.60)[a] |
| $I/\sigma I$ | 39.9 (7.2) |
| $R_{merge}$ (%) | 4.7 (22.6) |
| Completeness (%) | 99.7 (100.0) |
| Redundancy | 13.0 (10.5) |
| Mosaicity (°) | 0.20 |
| Refinement | |
| No. of reflections | 151866 |
| $R_{work}/R_{free}$ | 0.181/0.203 |
| No. of atoms | |
| Protein | 15875 |
| Ligand/Ion | 416 |
| Water | 448 |
| $B$-factor | |
| Protein | 47.4 |
| Ligand/Ion | 74.0 |
| Water | 50.0 |
| r.m.s. deviations [b] | |
| Bond lengths (Å) | 0.008 |
| Bond angles (°) | 1.07 |
| Ramachandran plot | |
| Most favored (%) | 97.1 |
| Additional allowed (%) | 2.8 |
| Generously allowed (%) | 0.0 |
| Disallowed (%) | 0.1 |
| Protein Data Bank Code | 8YH1 |

[a]Values of the highest resolution shells.

[b]Root mean square deviations.

the AlphaFold2 predicted [22] as a search model. Table 1 summarizes the statistics for the data collection and model refinement.

The crystal belongs to space group $I_{222}$ and contains nine ttUMPK molecules (chains A–I) per asymmetric unit (Fig 1a). The final structure was refined to 2.6 Å with $R_{work}$ and $R_{free}$ values of 18.1 and 20.3%, respectively. Size-exclusion analysis

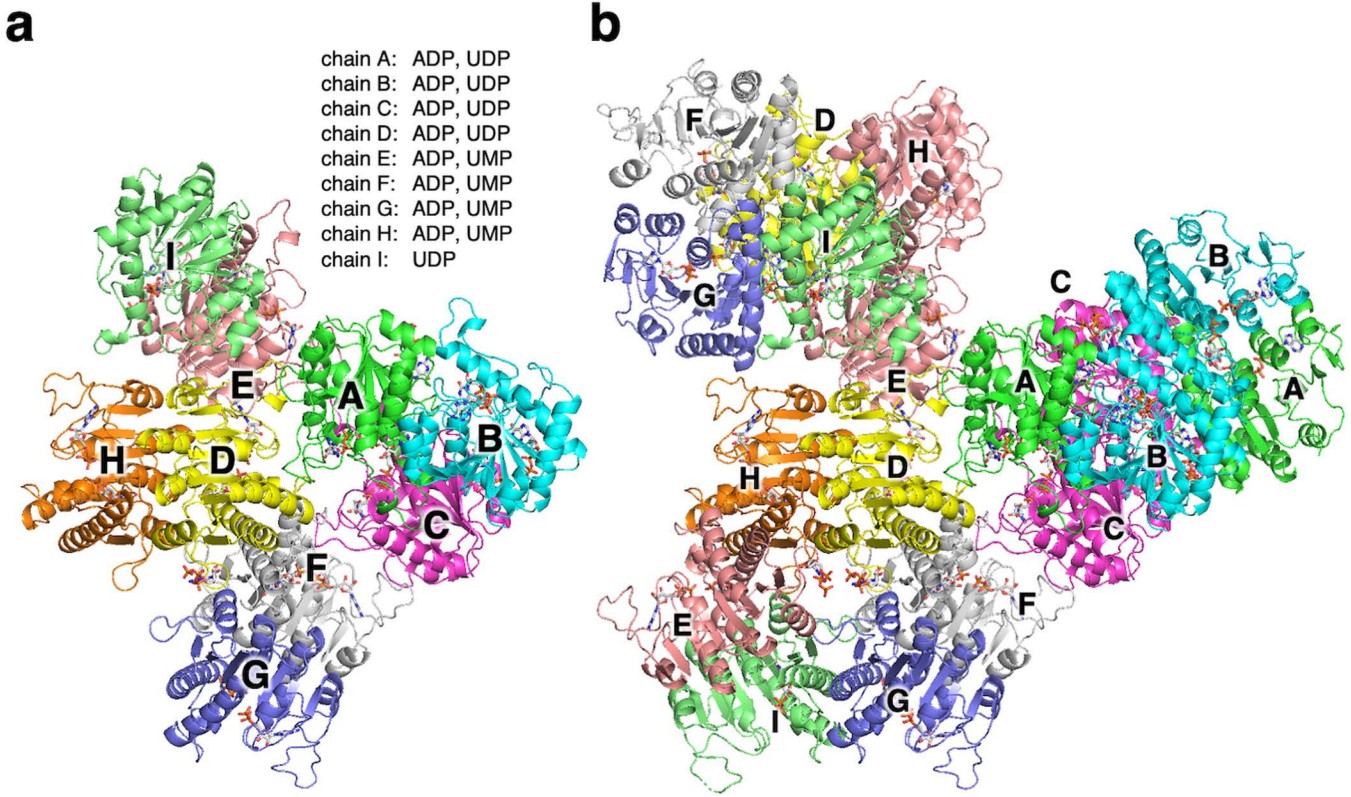

**Fig 1. Structure and assembly of ttUMPK. a** Nine chains in the asymmetric unit of the ttUMPK crystal structure. Chains A, B, C, D, E, F, G, H and I are colored green, cyan, magenta, yellow, dark pink, gray, blue, orange and pale green, respectively. Ligands bound to respective chains are indicated above the figure. **b** Hexameric assemblies generated by crystallographic symmetry operations. The colors of chains are the same as in **a**.

showed that ttUMPK forms a hexamer. We then applied crystallographic symmetry operations to the asymmetric unit to yield three hexameric forms as shown in Fig 1b. One hexamer comprises chains A, B and C, whereas the other two comprise chains D, E, F, G, H and I. The hexamer has the architecture of a trimer of dimers (Fig 2a), which is common to other bacterial UMPKs.

The subunit of ttUMPK has an α/β-fold with a eight-stranded β-sheet surrounded by eight α-helices and one $3_{10}$ helix (Fig 2b). The structures of the nine chains in the asymmetric unit are almost identical (Fig 2c). The loop regions are located at almost identical positions in all chains, but the loops of residues 156–179 (β5-β6 loop) show relatively high B-factors (Fig 2d). The region of residues 167–173 in chain I is disordered. The determined structure confirms that ttUMPK does not resemble CMP kinase and AMP kinase, and shows that its main-chain fold conforms to the amino acid kinase fold [30].

The ligand omit map shows residual electron density at the active site in the respective chains (Fig 3a). Upon crystallization, we added ADP, UDP and $MgCl_2$ to the protein solution. However, there are regions of the electron density map that are poorly defined in several chains, probably owing to low occupancy. As a result, we modeled various ligands in respective active sites by best fit with the electron density difference maps (Fig 1a). This structure represents the first report of the tertiary complex of bacterial UMPK with two substrates. In addition to ADP and UDP, the electron density for the small molecule is present at the central cavity of the hexamer in most chains (Fig 3b). The size and shape of this unidentified

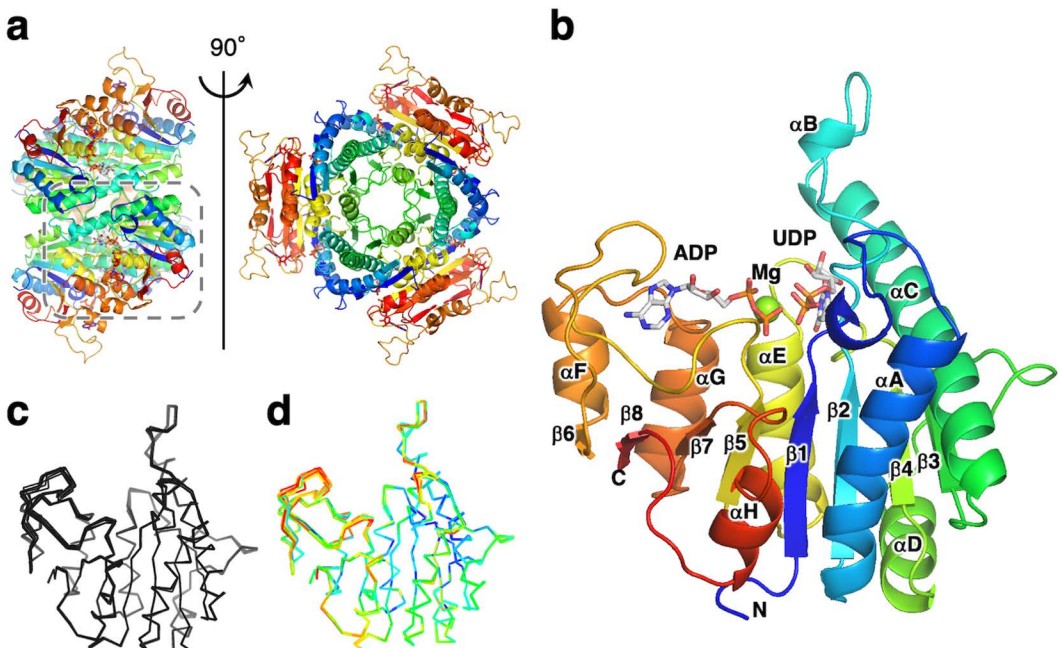

**Fig 2. Crystal structure of ttUMPK. a** Architecture of the hexameric organization. The subunits colored green, cyan and magenta indicate chains A, B and C (Fig 1), respectively. The view shown in the right panel represents a 90° rotation from that in the left panel. **b** Structure of the subunit. Chain A is represented. The chain is color ramped from blue (N-terminus) to red (C-terminus). Bound nucleotides are shown in a stick model. The Mg$^{2+}$ ion is shown as a sphere. **c** Superimposition of the backbones of all chains A–I. **d** B-factor diagram of all chains superimposed. The B-factor values are illustrated by color, ranging from low (blue) to high (red).

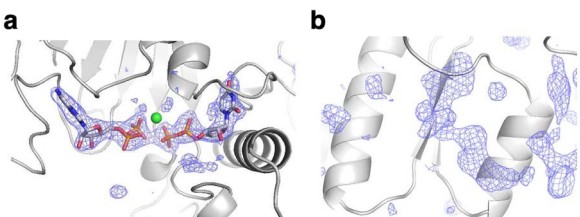

**Fig 3. The Fo-Fc omit electron density map of bound ligands.** a Electron density of the active-site region. b Electron density at the trimer interface. Bound ADP and UDP molecules (a) are shown as a stick model. The ligand-omit maps are shown at a contour level of 3 σ.

density are similar to a nucleotide. This will be discussed later. In the following, we mainly focused on the structure of chain A because it binds ADP, UDP and a magnesium ion.

For reference, we compared the crystal structure of ttUMPK with the one predicted by AlphaFold2 or SWISS-MODEL (Fig 4). The determined structure is more similar to the structure predicted by AlphaFold2 than by SWISS-MODEL, especially in several loop regions, β2–αB, αF–αG and β7–αH (encircled by a magenta dashed line in Fig 4). This accuracy of prediction seems important for molecular replacement to succeed.

## Active site

In the determined structure, ADP is located between the αF-αG and β4-β5 loops (Fig 5), as is common among archaeal UMPK structures. The β-phosphate moiety interacts with the side chains of Lys9 (3.0 Å), Ser11 (2.9 Å) and Lys157 (2.7 Å),

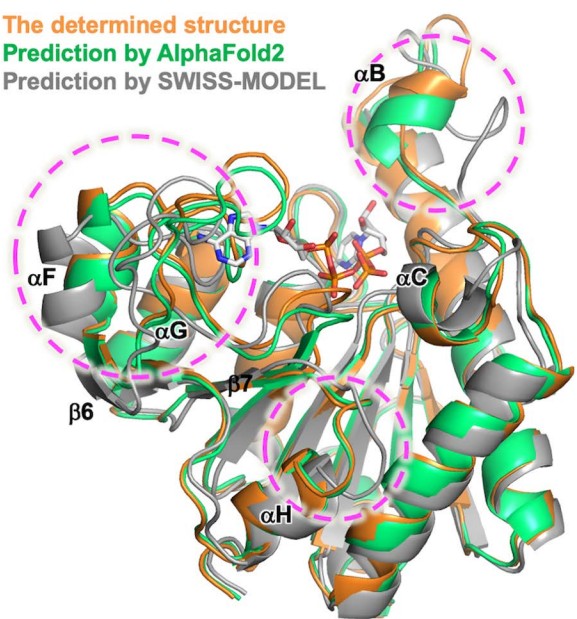

**Fig 4. Comparison of the crystal structure with the predicted structures of ttUMPK.** Superimposed structures of the determined structure, AlphaFold2- and SWISS-MODEL-predicted structures are colored orange, green and gray, respectively.

the main-chain amide of Asn158 (3.0Å) and a magnesium ion (2.5 Å). The 2'- and 3'-hydroxy groups of the ribose moiety form hydrogen bonds with the side chain of Asp167 (2.9 Å) and the main-chain carbonyl group of Asn158 (2.7 Å). The adenine moiety interacts with the main-chain amide and carbonyl of Tyr164 (2.7 and 3.0 Å).

UDP is located among the β1-αA, β2-αC and β4-αE loops (Fig 2b), as is common among other bacterial UMPK structures. The β-phosphate moiety interacts with the side chain of Lys9 (3.2 Å), Ser11 (3.3 Å) and the main-chain amide of Gly51 (2.8 Å) (Fig 5). The α-phosphate moiety interacts with the side chains of Ser136 (3.3 Å) and Thr137 (2.4 Å). Arg55 might interact with the β-phosphate of UDP (5.0 Å and 4.7 Å in chains C and I, respectively). The bridging oxygen atoms of the phosphate group are near the main-chain amide of Ala50 (3.0 and 3.3 Å). The 2'-hydroxy group of the ribose moiety forms a hydrogen bond with Asp69 (2.8 Å). The uracil moiety interacts with the side chains of Thr130 (3.1 Å) and Asp69 (3.7 Å) as well as main-chain carbonyl of Asn132 (3.1 Å) and Phe135 (3.0 Å).

The magnesium ion is coordinated by four oxygen atoms from the α- and β-phosphates of ADP as well as the β-phosphate of UDP and one water molecule (2.9 Å), and it may be coordinated by the side chain of Ser136 (3.4 Å). Lys9, Ser11 and probably Arg55 bridge ADP and UDP via interaction with the respective oxygen atoms. In chain G, the side chain of Arg55 is closer to both nucleotides (3.6 and 5.2 Å) than in chain A (~6.2 Å). It should be noted that the αB region around Arg55 has relatively a higher B-factor (Fig 2d).

Among these residues, Lys9, Ser11, Gly51, Arg55, Asp69, Thr130, Thr137, Tyr164 and Asp167 are highly conserved among UMPKs (Fig 6). It was reported that mutation of Arg62, Asp69 and Asp174 of *E. coli* UMPK (ecUMPK), corresponding to Arg55, Asp69 and Asp167 of ttUMPK, resulted in a substantial decrease in activity [10].

### Subunit interfaces

Two interfaces contribute to the packing between neighboring subunits in the ttUMPK hexamer (Fig 7a), the architecture of which is fundamentally the same as that of other prokaryotic homologues [4,5,7,9,13,14]. One interaface is mainly formed by polar and hydrophobic interactions interactions among αF helices, αG helices and β4-αE loops of adjacent

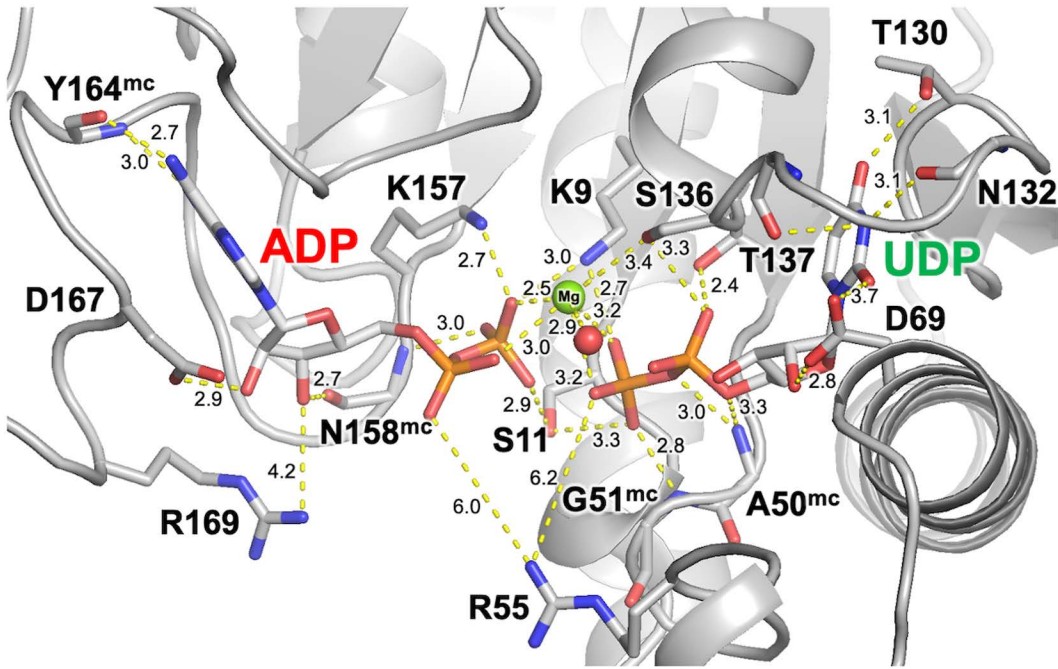

**Fig 5. Active site in the ADP-UDP complex.** Chain A is represented. Possible interactions are shown as dashed lines. The side and main chains are shown in a stick model. The Mg2+ ion and a water molecule are shown as a green and red sphere. The numbers are the distances shown in Å. Superscript mc represents the residues whose main-chain amide groups are involved in ligand recognition.

subunits (Figs 7b and 7c). At this interface, Asn132, Phe134, Phe135, Leu143 and Leu200 are well conserved (Fig 6). The other interface is formed by the hydrophobic interactions among αC helices, β2-αB loops and β3-αD loops (Figs 7d and 7e). Phe20, Ile22, Tyr60, Leu74, Leu81 and Val105 are well conserved (Fig 6). These extensive intersubunit interactions are supposed to stabilize the hexameric structure.

## Conformational changes upon substrate binding

To examine whether a conformational change occurs upon the binding of two substrate molecules, we compared the determined structure of the ADP-UDP–bound ttUMPK with those of the ligand-free and UDP-bound forms of *E. coli* UMPK (ecUMPK). It should be noted here that the GTP-bound form of ecUMPK is referred to as the "ligand-free" form because its active site is devoid of bound nucleotide. The overall structure of ttUMPK can be superimposed well with that of ecUMPK (Fig 8a). The anti-conformations of bound UDP molecules are common in both structures. Most residues involved in recognition of the phosphate moieties are conserved in these two structures. Lys9, Ser11, Gly51, Arg55, Asp69, Thr130, Thr137 and Asp167 in ttUMPK correspond to Lys15, Ser17, Gly58, Arg62, Asp77, Thr138, Thr145 and Asp174 in ecUMPK, respectively.

However, a striking difference among the three structures can be found in the β5-β6 loop. Upon the binding of UDP to ecUMPK, the β5-β6 loop shifts toward the active site even though this loop has no contact with UDP [13] (Fig 8a, orange). The $C_\alpha$ atom of Lys177, at the tip of the loop, shifts 6.1 Å toward the active site. Furthermore, in the UDP-ADP form of ttUMPK, the corresponding loop is closer to the active site than the UDP form of ecUMPK (Fig 8a, magenta). The $C_\alpha$ atom of Lys170 of ttUMPK, corresponding to Lys177 of ecUMPK, shifts 4.0 Å compared with the UDP-bound ecUMPK. This motion enables Asp167, Arg169 and Tyr164 to interact with the bound ADP (Fig 5). In archaeal UMPKs complexed with ATP analogues, the corresponding residues, Asp149, Lys171 and Tyr146 in *P. furiosus* UMPK (pfUMPK), interact with the bound adenine nucleotide similarly [4]. As for bacterial UDPK, the crystal structure of *B. anthracis* UDPK was reported to

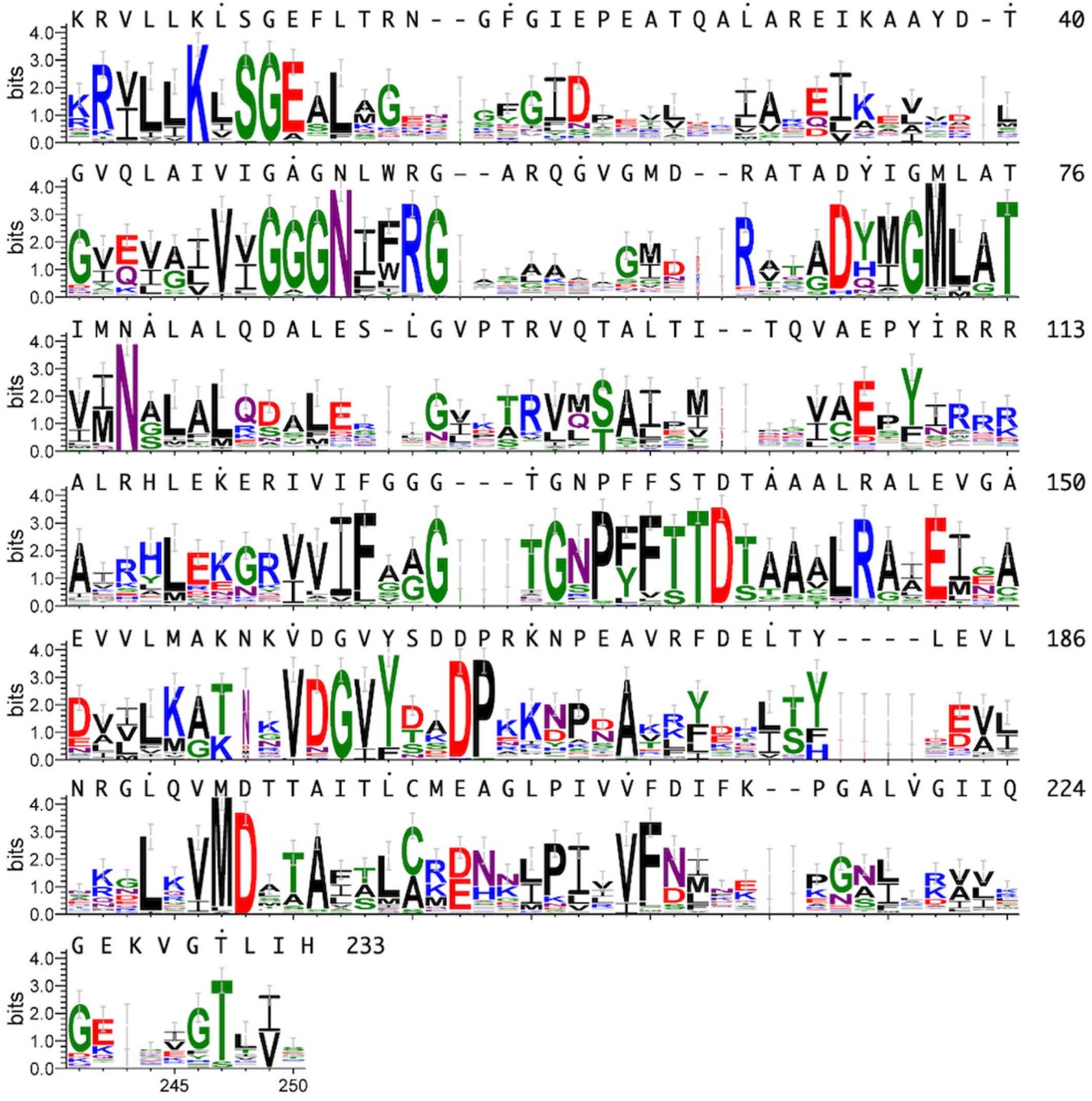

**Fig 6. Sequence conservation of bacterial UMPKs.** Sequence logo were generated with Weblogo3 [25] for the 50 most diverse members of AAK_UMPK-PyrH-Ec in the Conserved Domain Database [26].

be complexed with ATP, but the binding in the active site was believed to be in a nonproductive mode [9]. We have to consider carefully the comparison of UMPK structures derived from different species. Nevertheless, it can be assumed that in bacterial UMPKs the β5-β6 loop shifts toward the active site upon substrate binding.

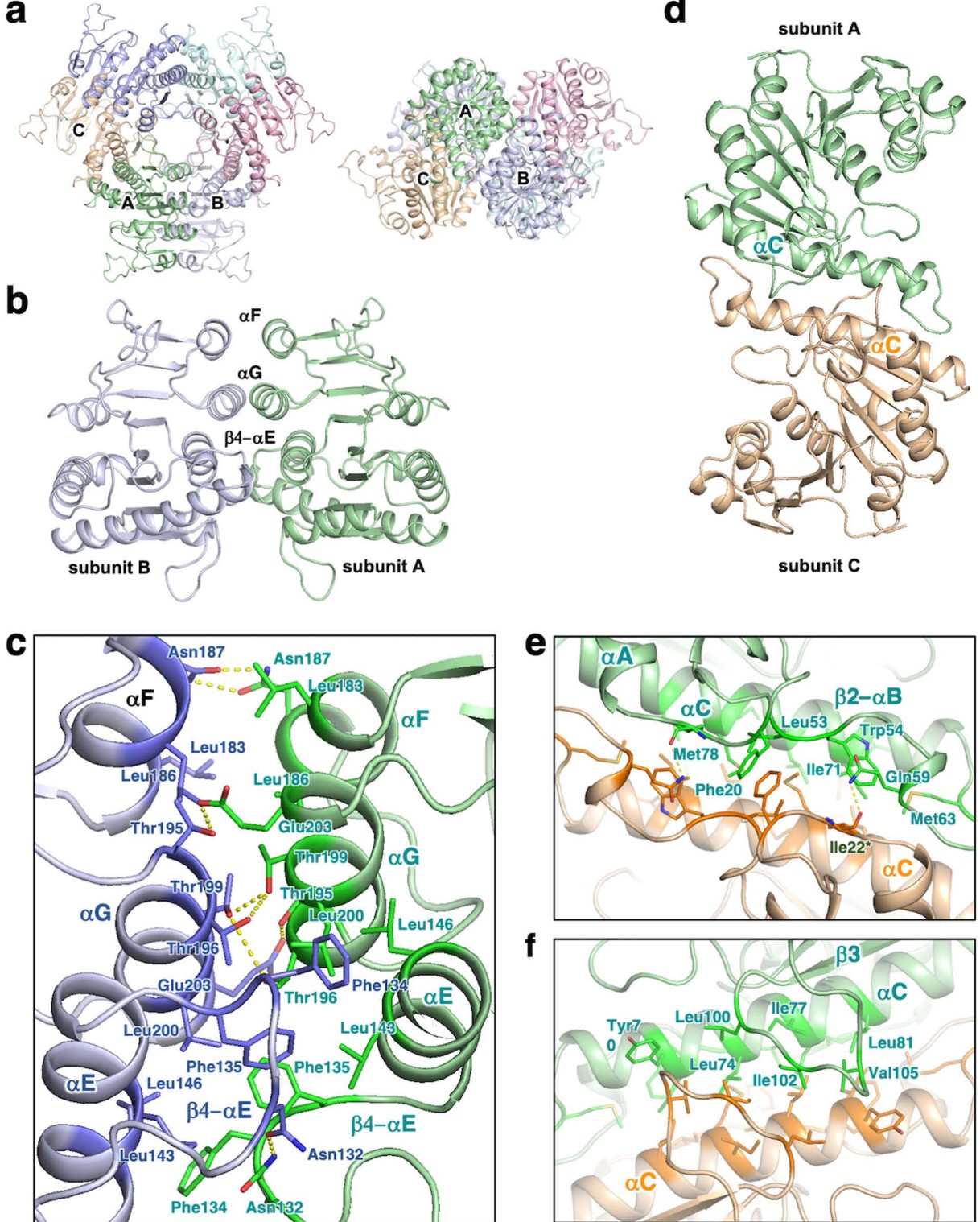

**Fig 7. Architecture of the hexamer organization of ttUMPK.** a Views from two sides. The subunits are colored differently. b Interaction between subunit A and B shown in panel a. This view is from the opposite side to that of panel a. c Expanded view of the subunit A–B interface. d Interaction between subunit A and C shown in panel a. e and f Expanded views of the subunit A–C interface. The view of panel f is from the opposite side to that of panel e. In panels c, e and f, the side chains thought to be involved in the interactions are shown as sticks and labeled.

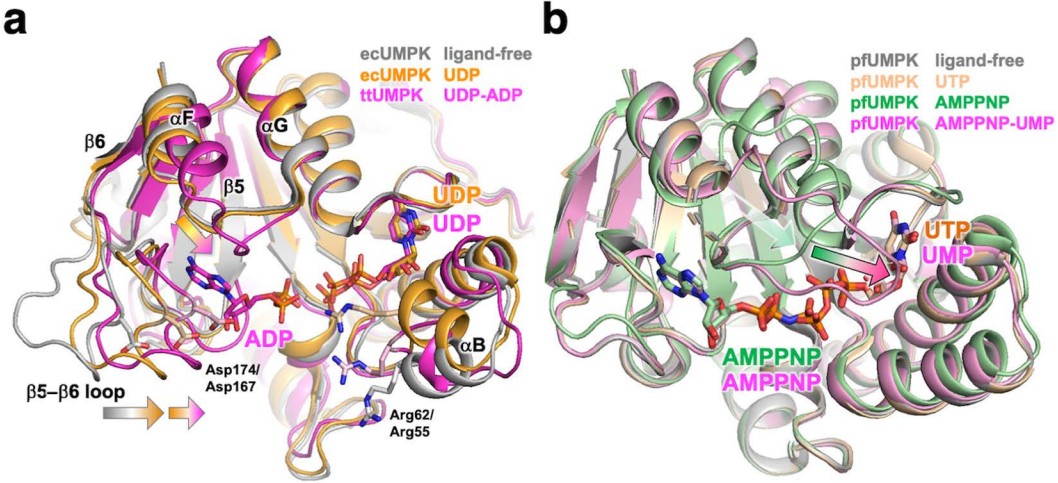

**Fig 8. Structural comparison of UMPKs in different liganded states.** a Bacteria UMPKs. Superimposed structures of ligand-free (PDB code 2V4Y) and UDP-bound (2BND) ecUMPK as well as ADP-UDP–bound ttUMPK (8YH1) are colored gray, orange and magenta, respectively. The side chains of Asp62/Asp55 and Asp174/Asp167 (ecUMPP/ttUMPK) are shown as stick models. b Archaeal UMPKs. Superimposed structures of ligand-free (PDB code 2BRX) and UTP- (2JI5), AMPPNP- (2BRI) and AMPPNP-UMP-bound (2BMU) pfUMPK are colored gray, light orange, green and magenta, respectively.

The proposed large shift of the β5-β6 loop is specific to bacterial UMPKs. Such a shift is not observed among archaeal UMPKs (Fig 8b). Differences in conformational changes might be related to the thermophilicity of the organisms from which the enzymes were derived. Structural comparison of several enzymes from thermophiles and mesophiles has suggested that the more thermostable the enzyme is, the smaller its motion is upon ligand binding [31]. This hypothesis may apply to UMPKs, because the large fluctuation may cause them to become unstable.

In archaeal UMPKs, another large conformational change is observed in different regions. In pfUMPK, the crystal structures have been determined for four states: ligand-free, UTP, AMPPNP and AMPPNP-UMP forms [4]. These structures can be superimposed very well (Fig 8b). The ligand-free and UTP-bound forms are almost identical to each other except for bound UTP, but the αF-αG loops are partially disordered in both forms. This loop becomes structured in the AMPPNP-bound form (Fig 8b, green). Furthermore, a large shift occurs in this αF-αG loop in the AMPPNP-UMP complex (Fig 8b, magenta). The long stretch corresponding to this loop region is specific to archaeal UMPKs, indicating that the large shift of this loop is specific to archaeal enzymes. The αF-αG loop of bacterial UMPK is shorter than that of pfUMPK, but the loop of ttUMPK shifts toward the bound ADP compared with ecUMPK up to 3.4 Å (Fig 8a, magenta), although no residue interacts with the nucleotide. The αF and β6 also move slightly in the same direction. These small shifts seem to be accompanied by the large shift of the β5-β6 loop described above.

The αB in ecUMPK also moves toward bound UDP (Fig 8a, orange) compared to the ligand-free form (Fig 8a gray). This motion prompts an interaction of Arg62, corresponding to Arg55 of ttUMPK, with the β-phosphate: the binding of UDP changes the positions of the $C_\alpha$ atom of Arg62 by 5.0 Å. The corresponding region in ttUMPK is situated between them in the ligand-free and UDP-bound forms (Fig 8a, magenta). Furthermore, this αB region has relatively high B-factor, suggesting greater flexibility (Fig 2d). The side chains of Arg55 have different conformations in the nine chains: this side chain is directed toward the bound substrates in chains E and G but not in the other chains. Therefore, it seems difficult to consider that the fluctuation in the αB region observed in these three structures represents a conformational change. The corresponding arginine residues, corresponding to Arg49 in pfUMPK, are located at similar positions in all the determined structures, suggesting no conformational change in archaeal UMPKs.

These results indicate a difference in conformational change upon substrate binding between bacterial and archaeal UMPKs. In both enzymes, long loops locally move toward the active site, albeit in distinct regions. This is quite different from the case of other NMP kinases, in which open-closed domain motion occurs upon substrate binding [1,18].

In addition, the conformational change observed for ttUMPK has similarities to that for other enzymes in the amino acid kinase family, to which UMPK belongs. Substrate binding in N-acetyl-L-glutamate kinase (NAGK), the representative enzyme of this family, is similar to that in ttUMPK. The ligand-free and N-acetyl-L-glutamyl phosphate-bound forms have almost the same structure (Fig 9, gray and green). Upon binding of AMPPNP and N-acetyl-L-glutamate, the loop (β11-β14) corresponding to the β5-β6 in ttUMPK moves toward the active site (Fig 8, orange). This movement is partly similar to that in ttUMPK. Moreover, the αF-αG and the connecting loop shift to make room for AMPPNP. The shift of the helices is relatively small in ttUMPK. Nevertheless, both enzymes seem common in the shift of the structural elements (β5-loop-β6-αF-loop-αG in ttUMPK) around the ATP-binding site upon substrate binding. Detailed analysis has suggested the importance of the intrinsic dynamics of NAGK and UMPK in their function expression [32,33].

## Allosteric site

Upon crystallization, we added only ADP and UDP as ligands; the ligand-omit map shows weak residual electron density at the trimer interface (Fig 3b), far from the active site. Based on previous knowledge of other UMPK structures [7,8,13,14], we speculated that this density could correspond to bound GTP. To examine whether GTP acts as an allosteric effector for the ttUMPK activity, like other bacterial UMPKs [34], we performed an enzyme assay of ttUMPK (Fig 10). The presence of GTP increased the $k_{cat}$ value 1.15-fold (13–15 s$^{-1}$) and the $K_m$ value for UMP 1.14-fold (46–56 µM). Because the activation was not pronounced, it was difficult to verify the binding of GTP. Nevertheless, in the following, we assumed that the unidentified molecule-bound site is the allosteric site in ttUMPK. The main reason for this assumption is that this unidentified density matches the position of GTP, the allosteric effector, in bacterial UMPKs from *H. pylori* [7], *M. tuberculosis* [8] and *X. campestris* [14].

During this study, we were able to collect another data set for ttUMPK and determined the crystal structure at 2.5 Å resolution. The $R_{work}$ and $R_{free}$ values are 19.6% and 24.3%, respectively. The arrangement of nine chains in the asymmetric unit is fundamentally the same as that described above, but most of the bound uridine nucleotides in the active site are UMP, not UDP. In addition, no electron density is observed at the trimer interface, that is, the allosteric site. Assuming that this structure represents

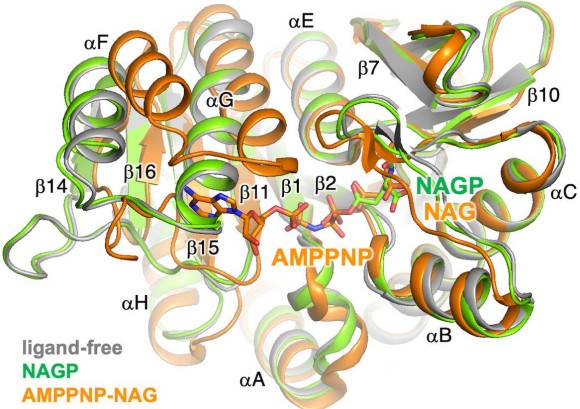

**Fig 9. Structural comparison of NAGK in various liganded states.** Superimposed structures of ligand-free, N-acetyl-L-glutamyl phosphate (NAGP)-bound and AMPPNP-N-acetylglutamate (NAG)-bound NAGK are colored gray, green and orange, respectively. Their PDB codes are 2WXB, 2X2W and 1GS5. Secondary structure elements are labeled according to the nomenclature of Ramón-Maiques [30].

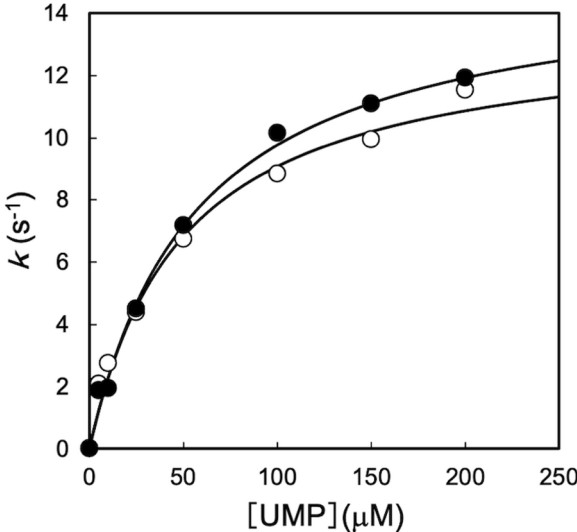

**Fig 10. Phosphorylation activity of ttUMPK.** The activity was assayed as described under "Materials and methods." The ordinate (k) represents the apparent rate constant. The solid lines represent the theoretical curves obtained by fitting the data to the Michaelis-Menten equation. The symbols are the following: open circles, without GTP; closed circles, with 0.5 mM GTP.

an effector-free form, it is unlikely that a significant conformational change occurs in ttUMPK upon binding of an allosteric effector. This seems consistent with the results of the structural analysis of UMPKs from *E. coli* [13], *H. pylori* [7] and *X. campestris* [14]. It should be mentioned that in these UMPKs slight changes are found at the ATP-binding site, not the GTP-binding site, upon GTP binding. It is also suggested that GTP binding makes the conformation more compact for *Streptococcus pneumonia* UMPK [35]. However, these reported differences are not enough to explain the allosteric mechanism by GTP. Further studies are required to understand molecular mechanisms underlying the complexity of the allosteric regulation in UMPK.

## Acknowledgments

This work was supported in part by the RIKEN Structural Genomics/Proteomics Initiative and the National Project on Protein Structural and Functional Analyses, Ministry of Education, Culture, Sports, Science and Technology of Japan.

## Author contributions

**Conceptualization:** Noriko Nakagawa, Seiki Kuramitsu, Ryoji Masui.

**Data curation:** Kenji Fukui, Noriko Nakagawa, Ryoji Masui.

**Investigation:** Kenji Fukui, Anzu Nishiwaki, Noriko Nakagawa, Ryoji Masui.

**Methodology:** Kenji Fukui, Noriko Nakagawa.

**Project administration:** Ryoji Masui.

**Supervision:** Kenji Fukui, Seiki Kuramitsu, Ryoji Masui.

**Validation:** Kenji Fukui, Noriko Nakagawa, Ryoji Masui.

**Visualization:** Kenji Fukui, Anzu Nishiwaki.

**Writing – original draft:** Kenji Fukui, Ryoji Masui.

**Writing – review & editing:** Kenji Fukui, Anzu Nishiwaki, Noriko Nakagawa, Seiki Kuramitsu, Ryoji Masui.

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
