## [Decision Letter · Decision Letter 0]

10 Jun 2025

Dear Dr. Masui,

We look forward to receiving your revised manuscript.

Kind regards,

Michael Massiah

Academic Editor

PLOS ONE

Journal Requirements:

Reviewers' comments:

Reviewer's Responses to Questions

**Comments to the Author**

1. Is the manuscript technically sound, and do the data support the conclusions?

Reviewer #1: Partly

Reviewer #2: Yes

2. Has the statistical analysis been performed appropriately and rigorously?

Reviewer #1: N/A

Reviewer #2: N/A

3. Have the authors made all data underlying the findings in their manuscript fully available?

Reviewer #1: Yes

Reviewer #2: Yes

4. Is the manuscript presented in an intelligible fashion and written in standard English?

Reviewer #1: Yes

Reviewer #2: Yes

Reviewer #1: The manuscript "The crystal structure of Thermus thermophilus UMP kinase complexed with a

phosphoryl group acceptor and donor" by Fukui, K et al. describes and compares the crystal structure of ttUMPK to other bacterial and archaeal UMPK structures and reveals a loop movement that contains for the first time a UMPK with both a phosphate group donor and acceptor.

Overall, I found the manuscript to be well written with some minor flaws that could be readily fixed but lacks scientific depth which is probably not the authors fault. For example, when comparing movement or sequence it would be nice to have the relevance, ie. is the movement important for activity or stability, does the ligand affinity increase or decrease. These key residues can either be tested or referenced from other structures or manuscripts. Also, sequence based alignment of said residues with other species and the relevance, the authors have included structure based alignment. The structure of ttUMPK is a trimer of dimers, the authors should address whether the interface is conserved and potentially perform mutagenesis to disrupt the interactions which should then be confirmed by SEC and functional assays, this data should then be included in the manuscript. It is worth noting that the authors have included in the manuscript the description of a functional assay for ttUMPK but I only found functional data for Kcat and Km on lines 311-312 of the manuscript with no accompanying figure.

Reviewer #2: The present publication describes a ternary complex of tturidine monophosphate kinase (UMPK) with a phosphoryl donor and substrate. Upon comparison with other bacterial UMPK structures, the authors identified conformational changes that occur upon substrate and co-substrate binding. Specifically, a very drastic conformational change is observed at the b5-b6 loop upon ADP-UDP binding, which is also seen upon substrate binding by the amino acid kinases family, to which ttUMPK belongs. This conformational change is not as pronounced in archaeal UMPK. This novel structure serves as a missing piece for proposing a model for conformational changes upon substrate binding in thermophilic UMPKs, thereby expanding the knowledge within this field.

I recommend this article for publication with minor revisions:

1. The authors propose that the observed positive electron density at the trimer interface corresponds to GTP, based on previously published structures of UMPKs. However, there is no direct evidence that GTP binds to ttUMPK. Seeing as GTP was not added to the crystallization conditions and that GTP did not have significant effect on UMPK’s enzyme kinetics, one can only speculate on the identity of the observed trimer interface density. Having GTP built into the structure could be misleading.

2. Line 23: “addition, GTP, an allosteric effector, bound to intersubunit cavity in the center of the” This statement seems very definitive, when there is not concrete evidence that GTP is indeed bound to this structure. Binding studies (e.g. ITC) could perhaps address the question of GTP binding. I don’t think binding studies are necessary for publication, but the language should be modified to a more speculatory tone.

3. Line 44: Unclear what is a substrate, analogue, phosphoryl acceptor

4. Oligomeric state of ttUMPK. Have the authors attempted other approaches to determining the oligomeric state of ttUMPK. Techniques like Analytical Ultracentrifugation, Circula Dichroism, and/or SEC-multi angle light scattering are better determinants of oligomeric state than Size Exclusion is. Then again, I do not think this impacts the overall findings from this article.

5. Were all chains containing the substrate and co-substrate compared? I am curious to know if other chains within the AU exhibited similar conformational changes.

6. Label helices and loops in Figure 7

7. Can the activity data be included as a supplement? It might be worth comparing the determined kinetic parameters to those of bacterial UMPK +/- GTP

8. Was the “GTP-free” structure (described in paragraph starting in line 336) treated/prepared differently from the one described published in this study?

**Do you want your identity to be public for this peer review?** For information about this choice, including consent withdrawal, please see our Privacy Policy

Reviewer #1: No

Reviewer #2: No

---

## [Author Response · Author response to Decision Letter 1]

25 Jun 2025

We found the reviewers’ comments most helpful. We revised the manuscript with considering reviewers' comments as follows. The texts after ==> are our comments on the revision.

Review Comments to the Author

Reviewer #1:

The manuscript "The crystal structure of Thermus thermophilus UMP kinase complexed with a phosphoryl group acceptor and donor" by Fukui, K et al. describes and compares the crystal structure of ttUMPK to other bacterial and archaeal UMPK structures and reveals a loop movement that contains for the first time a UMPK with both a phosphate group donor and acceptor.

Overall, I found the manuscript to be well written with some minor flaws that could be readily fixed but lacks scientific depth which is probably not the authors fault. For example, when comparing movement or sequence it would be nice to have the relevance, ie. is the movement important for activity or stability, does the ligand affinity increase or decrease. These key residues can either be tested or referenced from other structures or manuscripts.

==> According to the reviewer's comment, we cited previous studies on mutational analysis of E. coli UMPK (Bucurenci et al., 1997, J. Bacteriol. 180, 473-477) and added the sentence as follows: It was reported that mutation of Arg62, Asp69 and Asp174of E. coli UMPK (ecUMPK), corresponding to Arg55, Asp69 and Asp167 of ttUMPK, resulted in a substantial decrease in activity. We also found a significant decrease in kcat was observed in the Ala- and Glu-substituted mutants of ttUMPK (manuscript in preparation).

Also, sequence based alignment of said residues with other species and the relevance, the authors have included structure based alignment.

==> According to the reviewer's comment, we created the sequence logo for the 50 most diverse members of AAK_UMPK-PyrH-Ec (bacterial UMPKs) in the Conserved Domain Database. We think the sequence logo is a better way to examine conservation than sequence alignment. The logo is shown as Figure 6. In addition, we described the method to create sequence logo in the new section "Sequence and structural data analysis" in Materials and Methods as follows: Sequence logos were generated with Weblogo3 for the 50 most diverse members of AAK_UMPK-PyrH-Ec in the Conserved Domain Database (p. 8).

The structure of ttUMPK is a trimer of dimers, the authors should address whether the interface is conserved and potentially perform mutagenesis to disrupt the interactions which should then be confirmed by SEC and functional assays, this data should then be included in the manuscript.

==> According to the reviewer's comment, we added the description of the structural details of the subunit interfaces in the new section "Subunit interfaces" (p. 12–13). We made the new figure showing the details of the two interfaces as Figure 7. This revealed that the hydrophobic residues are frequent in the interface. Some of these residues are well conserved, as shown in Figure 6 (the sequence logo). Since many hydrophobic residues are likely to be involved in intersubunit intaraction, it is difficult to predict which residues are critical for oligomerization. It also seems unlikely that site-directed mutation at one or two sites leads to changes in the oligomeric state. Therefore, we have not performed experiments investigating this interaction.

It is worth noting that the authors have included in the manuscript the description of a functional assay for ttUMPK but I only found functional data for Kcat and Km on lines 311-312 of the manuscript with no accompanying figure.

==> According to the reviewer's comment, we added a new figure showing the results of enzyme assay of ttUMPK as Figure 10.

Reviewer #2:

The present publication describes a ternary complex of tturidine monophosphate kinase (UMPK) with a phosphoryl donor and substrate. Upon comparison with other bacterial UMPK structures, the authors identified conformational changes that occur upon substrate and co-substrate binding. Specifically, a very drastic conformational change is observed at the b5-b6 loop upon ADP-UDP binding, which is also seen upon substrate binding by the amino acid kinases family, to which ttUMPK belongs. This conformational change is not as pronounced in archaeal UMPK. This novel structure serves as a missing piece for proposing a model for conformational changes upon substrate binding in thermophilic UMPKs, thereby expanding the knowledge within this field.

I recommend this article for publication with minor revisions:

1. The authors propose that the observed positive electron density at the trimer interface corresponds to GTP, based on previously published structures of UMPKs. However, there is no direct evidence that GTP binds to ttUMPK. Seeing as GTP was not added to the crystallization conditions and that GTP did not have significant effect on UMPK’s enzyme kinetics, one can only speculate on the identity of the observed trimer interface density. Having GTP built into the structure could be misleading.

==> According to the reviewer's comment, we deleted a GTP model from Figure 3b (density map) and removed GTP molecules from Figure 2. We also deleted (original) Figures 8 and 9, showing the binding mode of GTP in ttUMPK and its homologues, respectively, and subsequently deleted the texts regarding these figures. After removing GTP from the coordinate file, we performed the refinement again and re-created Figure 3 based on a new electron density map. We also revised the values in Table 1. We have completed the updating of the PDB entry (8YH1) and uploaded the validation reports file upon submitting the revised manuscript.

2. Line 23: “addition, GTP, an allosteric effector, bound to intersubunit cavity in the center of the” This statement seems very definitive, when there is not concrete evidence that GTP is indeed bound to this structure. Binding studies (e.g. ITC) could perhaps address the question of GTP binding. I don’t think binding studies are necessary for publication, but the language should be modified to a more speculatory tone.

==> According to the reviewer's comment, we deleted from Abstract the following sentence: In addition, GTP, an allosteric effector, bound to intersubunit cavity in the center of the hexametric structure.

3. Line 44: Unclear what is a substrate, analogue, phosphoryl acceptor

==> According to the reviewer's comment, we revised the two sentences that are pointed out as follows: In archaea, the crystal structures of UMPK complexed with UMP, UMP-AMPPNP and UTP have been determined for Pyrococcus furiosus [4]. The structures complexed with AMPPNP and UMP-AMPPNP have also been determined for Sulfolobus solfataricus [5]. (p. 3)

4. Oligomeric state of ttUMPK. Have the authors attempted other approaches to determining the oligomeric state of ttUMPK. Techniques like Analytical Ultracentrifugation, Circula Dichroism, and/or SEC-multi angle light scattering are better determinants of oligomeric state than Size Exclusion is. Then again, I do not think this impacts the overall findings from this article.

==> The SEC-MALS and analytical ultracentrifuge are not available to our labs. We employed dynamic light scattering (DLS) to characterize the approximate size of ttUMPK in solution, and the molecular mass was estimated to be 126-135 kDa, assuming a globular shape of the particle. The theoretical molecular mass of the hexameric ttUMPK is 152 kDa. However, this is a preliminary experiment, and we have not perrformed other additional experiments investigating the oligomeric state of ttUMPK.

5. Were all chains containing the substrate and co-substrate compared? I am curious to know if other chains within the AU exhibited similar conformational changes.

==> In the original manuscript, Figure 2c showed the superimposition of the backbones of all chains A–I, and we described as follows: The structure of the nine chains in the asymmetric unit are almost identical (Fig. 2c), but the loops of residues 156-179 (beta5-beta6 loop) show relatively high B-factors (Fig 2d). Following the reviewer's comment, we re-examined the results and found no differences between the chains, describing all nine chains as nearly identical.

6. Label helices and loops in Figure 7

==> According to the reviewer's comment, we added the labels for helices and strands based on the reference (Ramón-Maiques et al., 2002) and added the sentence to the legend to this figure (renamed to Figure 9): Secondary structure elements are labeled according to the nomenclature of Ramón-Maiques (p. 16). We also revised the phrase from "the loop corresponding to the beta5-beta6 in ttUMPK" to "the loop (beta11-beta14) corresponding to the beta5-beta6 in ttUMPK." (p. 16)

7. Can the activity data be included as a supplement? It might be worth comparing the determined kinetic parameters to those of bacterial UMPK +/- GTP

==> According to the reviewer's comment, we added a new figure showing the results of enzyme assay of ttUMPK as Figure 10. We cited this figure in the text (p. 17).

8. Was the “GTP-free” structure (described in paragraph starting in line 336) treated/prepared differently from the one described published in this study?

==> The diffraction data of the "GTP-free" structure was obtained by using different crystals from those in this study, but the protein solutions were originated from the same preparation. Differences in crystallization conditions may result in differences in the bound ligands. In the process of reviewing the data, we realized that we had made a mistake with the crystallization conditions. Therefore, we revised the description about the crystallization conditions (p. 5).

---

## [Decision Letter · Decision Letter 1]

23 Jul 2025

Dear Dr. Masui,

Thank you for submitting your manuscript to PLOS ONE. After careful consideration, we feel that it has merit for PLOS ONE’s publication criteria as it currently stands. While the original reviewers have accepted the revision, there was my comment in the original latter that wanted you to address the previous structure instead of giving the impression of this being the first and only structure. That was not addressed in the revision. 

We look forward to receiving your revised manuscript.

Kind regards,

Michael Massiah

Academic Editor

PLOS ONE

Journal Requirements:

Reviewers' comments:

Reviewer's Responses to Questions

**Comments to the Author**

Reviewer #1: All comments have been addressed

Reviewer #2: All comments have been addressed

2. Is the manuscript technically sound, and do the data support the conclusions?

Reviewer #1: Yes

Reviewer #2: Yes

3. Has the statistical analysis been performed appropriately and rigorously?

Reviewer #1: Yes

Reviewer #2: N/A

4. Have the authors made all data underlying the findings in their manuscript fully available?

Reviewer #1: Yes

Reviewer #2: Yes

5. Is the manuscript presented in an intelligible fashion and written in standard English?

Reviewer #1: Yes

Reviewer #2: Yes

Reviewer #1: I want to thank the authors' of the manuscript "The crystal structure of Thermus thermophilus UMP kinase complexed with a phosphoryl group acceptor and donor" by Fukui, K. et al. for taking the time and effort to review my comments and address my concerns. I am satisfied with the authors' revised manuscript and recommend it be accepted for publication.

Reviewer #2: (No Response)

**Do you want your identity to be public for this peer review?** For information about this choice, including consent withdrawal, please see our Privacy Policy

Reviewer #1: No

Reviewer #2: No

---

## [Author Response · Author response to Decision Letter 2]

30 Jul 2025

We found the academic editor's comments most helpful. We revised the manuscript with considering reviewers' comments as follows. The texts after ==> are our comments on the revision.

Editor:

Furthermore, I realized that I was the academic editor for your recent article on CMP and its complex for which there no mention of in this article. Thus, this article gives the impression that this is the first thermophilic NMPkinase solved with its substrates, which is misleading. It is important to note this upfront and why UMP structure is necessary (sequence comparision etc) and what is learned from your various structure of UMP and substrates vs CMP findings. Please ensure a discussion on the comparison of the two proteins as well.

==> As mentioned in Introduction section, bacterial UMPKs have no sequence similarity to other known NMP kinases, including CMP kinase, but have significant sequence and structural similarity to amino acid kinases. Therefore, in the original version of our manuscript, we did not mention our previous work about T. thermophilus CMPK (Mega et al. 2020). To avoid misleading, we added the following sentence in Introduction: Among T. thermophilus NMP kinases, the crystal structures of CMP kinase in various liganded states have been determined (Mega, 2020). The apo form structure of AMP kinase has also been determined (PDB ID, 3CM0) (p. 4).

In addition, to highlight that bacterial UMPK and CMPK belong to different families, we added the following sentence in Results and Discussion: The determined structure confirms that ttUMPK does not resemble CMP kinase and AMP kinase, and shows that its main-chain fold conforms to the amino acid kinase fold (Ramón-Maiques, 2002) (p. 9).

As descirbed above, bacterial UMPK and CMPK belong to different protein families. We believe that no useful information can be gained from the (structural) comparison of these two proteins. Therefore, I did not add any further comments about them.

---

## [Editor Report · Decision Letter 2]

1 Aug 2025

The crystal structure of Thermus thermophilus UMP kinase complexed with a phosphoryl group acceptor and donor

PONE-D-25-12007R2

Dear Dr. Masui,

We’re pleased to inform you that your manuscript has been judged scientifically suitable for publication and will be formally accepted for publication once it meets all outstanding technical requirements.

Kind regards,

Michael Massiah

Academic Editor

PLOS ONE

Additional Editor Comments (optional):

Thanks for addressing my comments in this revision.
---

## [Editor Report · Acceptance letter]

PONE-D-25-12007R2

PLOS ONE

Dear Dr. Masui,

I'm pleased to inform you that your manuscript has been deemed suitable for publication in PLOS ONE. Congratulations! Your manuscript is now being handed over to our production team.

Kind regards,

on behalf of

Dr. Michael Massiah

Academic Editor

PLOS ONE